# Association between use of systemic and inhaled glucocorticoids and changes in brain volume and white matter microstructure: a cross-sectional study using data from the UK Biobank

Merel van der Meulen  , Jorge Miguel Amaya, Olaf M Dekkers, Onno C Meijer

**To cite:** van der Meulen M, Amaya JM, Dekkers OM, *et al*. Association between use of systemic and inhaled glucocorticoids and changes in brain volume and white matter microstructure: a cross-sectional study using data from the UK Biobank. *BMJ Open* 2022;**12**:e062446. doi:10.1136/bmjopen-2022-062446

Department of Medicine, Division of Endocrinology, Leiden University Medical Center, Leiden, The Netherlands

**Correspondence to**
Dr Merel van der Meulen;
m.vandermeulen@lumc.nl

## ABSTRACT

**Objective** To test the hypothesis that systemic and inhaled glucocorticoid use is associated with changes in grey matter volume (GMV) and white matter microstructure.

**Design** Cross-sectional study.

**Setting** UK Biobank, a prospective population-based cohort study of adults recruited in the UK between 2006 and 2010.

**Participants** After exclusion based on neurological, psychiatric or endocrinological history, and use of psychotropic medication, 222 systemic glucocorticoid users, 557 inhaled glucocorticoid users and 24 106 controls with available T1 and diffusion MRI data were included.

**Main outcome measures** Primary outcomes were differences in 22 volumetric and 14 diffusion imaging parameters between glucocorticoid users and controls, determined using linear regression analyses adjusted for potential confounders. Secondary outcomes included cognitive functioning (six tests) and emotional symptoms (four questions).

**Results** Both systemic and inhaled glucocorticoid use were associated with reduced white matter integrity (lower fractional anisotropy (FA) and higher mean diffusivity (MD)) compared with controls, with larger effect sizes in systemic users (FA: adjusted mean difference (AMD)=−3.7e-3, 95% CI=−6.4e-3 to 1.0e-3; MD: AMD=7.2e-6, 95% CI=3.2e-6 to 1.1e-5) than inhaled users (FA: AMD=−2.3e-3, 95% CI=−4.0e-3 to −5.7e-4; MD: AMD=2.7e-6, 95% CI=1.7e-7 to 5.2e-6). Systemic use was also associated with larger caudate GMV (AMD=178.7 mm$^3$, 95% CI=82.2 to 275.0), while inhaled users had smaller amygdala GMV (AMD=−23.9 mm$^3$, 95% CI=−41.5 to −6.2) than controls. As for secondary outcomes, systemic users performed worse on the symbol digit substitution task (AMD=−0.17 SD, 95% CI=−0.34 to −0.01), and reported more depressive symptoms (OR=1.76, 95% CI=1.25 to 2.43), disinterest (OR=1.84, 95% CI=1.29 to 2.56), tenseness/restlessness (OR=1.78, 95% CI=1.29 to 2.41), and tiredness/lethargy (OR=1.90, 95% CI=1.45 to 2.50) compared with controls. Inhaled users only reported more tiredness/lethargy (OR=1.35, 95% CI=1.14 to 1.60).

## STRENGTHS AND LIMITATIONS OF THIS STUDY

⇒ To the best of our knowledge, this is the largest study to date assessing the association between glucocorticoid use and brain structure, and the first to investigate these associations in inhaled glucocorticoid users.

⇒ Relatively strict exclusion criteria were used to limit the potential confounding that may arise in observational cohort studies.

⇒ However, the cross-sectional nature of this study precludes formal conclusions on causality.

⇒ Dose and duration of medication use were not available in the UK Biobank, making thorough analyses on dose-dependent or duration-dependent associations impossible.

**Conclusions** Both systemic and inhaled glucocorticoid use are associated with decreased white matter integrity and limited changes in GMV. This association may contribute to the neuropsychiatric side effects of glucocorticoid medication, especially with chronic use.

## INTRODUCTION

Due to their immunosuppressive properties, glucocorticoids are among the most prescribed drugs on the market, with an estimated annual prevalence of systemic glucocorticoid use between 0.5% and 3%.[1–5] Although efficacious, both systemic and local (especially inhaled) glucocorticoids are associated with many potentially serious metabolic, cardiovascular and musculoskeletal side effects.[6–9] Besides these physical side effects, the use of synthetic glucocorticoids is also associated with neuropsychiatric symptoms and disorders, including depression, mania, delirium and even a sevenfold increased suicide (attempt) rate.[10 11] In addition, on an anatomical level, both preclinical and clinical studies have shown long-lasting effects of glucocorticoid overexposure on the brain. In

patients with chronic endogenous glucocorticoid excess due to a pituitary tumour (Cushing disease), it has been established that long-term glucocorticoid excess is associated with global cerebral atrophy[12–18] and decreased cortical thickness and grey matter volumes in specific brain regions.[13 18–26] Some of these effects were detected even after ten years of biochemical remission.[22 23] Moreover, a few small studies have shown volumetric reductions in specific brain regions, including the hippocampus and amygdala,[27–31] in patients using chronic and/or high-dose synthetic systemic glucocorticoids. Besides these structural abnormalities, several studies in animal models and patients with Cushing disease have also demonstrated widespread reductions in white matter integrity throughout the brain.[32–36] In humans, this was studied using diffusion tensor imaging (DTI), showing globally decreased fractional anisotropy (FA), which represents the directionality of water diffusion through the brain and is a marker of microstructural architecture,[37] and increased mean diffusivity (MD),[32–35] which represents an increase in water diffusion in all directions and is associated with disease processes such as inflammation and oedema.[37]

However, most clinical studies investigating the effects of glucocorticoid overexposure on brain structure have been performed in small, selected populations with chronic glucocorticoid excess due to Cushing disease or systemic glucocorticoid use. It remains unknown whether these associations can also be observed in a broader sample of people using glucocorticoids, including inhaled glucocorticoids. We, therefore, used data from the UK Biobank, a large population-based cohort study, to investigate whether, at a population level, differences in brain volumes and white matter microstructure could be detected between systemic or inhaled glucocorticoid users and non-users. As secondary outcomes, we also assessed potential differences in cognitive and emotional functioning. Based on previous literature, we hypothesised that glucocorticoid use would be associated with decreased grey matter volumes in the limbic system and hippocampus, a widespread reduction in FA and increase in MD throughout the brain, and poorer cognitive and emotional outcomes.

## METHODS

### Study design

The UK Biobank is a large population-based prospective cohort, comprising over 500 000 participants aged 40–69 years at the time of recruitment (between 2006 and 2010).[38]

### Data collection

Data were collected at the assessment centres and during an online follow-up. Data used for this study included data on demographic characteristics, health and medical history, brain imaging, cognitive and emotional functioning, and body composition. Data on demographic characteristics, cognition and emotional functioning were collected using a touch screen device at the assessment centres. If patients had indicated that they did not want to answer a question on one or more of these characteristics, we coded this as missing. Data on health and medical history, including medication use, were collected using the touch screen device and a verbal interview (self-reported data), but also using Hospital Episode Statistics (HES). Body composition was measured using body impedance on a Tanita BC418MA body composition analyser as described in the UK Biobank documentation.[39] The imaging acquisition is described in more detail below.

### Participants

For the analysis presented in this study, we selected participants who

1. Had both T1-weighted MRI and DTI data available at the same imaging visit.
2. Did not have a history of psychiatric disease based on self-reported data or HES data. However, we did include the psychiatric diseases most commonly associated with glucocorticoid use based on previous literature (anxiety, depression, mania and delirium)[10] as we did not want to exclude patients based on potentially glucocorticoid-related outcomes.
3. Did not use psychotropic medication.
4. And did not have any neurological condition based on self-reported or HES data.

Individuals who met these criteria and used oral or parenteral glucocorticoids at the time of imaging were included in the systemic glucocorticoid patient group (n=222), and individuals who met these criteria and used inhaled glucocorticoids (but no systemic glucocorticoids) at the time of imaging were included in the inhaled glucocorticoid group (n=557). Among the patients using systemic glucocorticoids, 14 were also using inhaled glucocorticoids. Individuals who met these criteria but had not used systemic or inhaled glucocorticoids at any time point (before and including the imaging visit) and did not have any endocrinological disorder according to self-reported or HES data, were included in the control group (n=24 106). A flowchart of patient selection is presented in figure 1, and online supplemental file 1 provides a list of all Biobank UK field codes that were used as inclusion or exclusion criteria.

### Imaging data

Our study made use of imaging-derived phenotypes (IDPs) generated by an image-processing pipeline developed and run on behalf of the UK Biobank. Details on the brain imaging acquisition protocols, imaging processing and quality control, and generation of IDPs are provided by the UK Biobank.[40 41] In short, all imaging was performed on a standard Siemens Skyra 3 Tesla scanner with a standard Siemens 32-channel radiofrequency receiver head coil. T1-weighted imaging was performed using a three-dimensional magnetisation-prepared rapid acquisition with gradient echo sequence (3D MPRAGE) in the

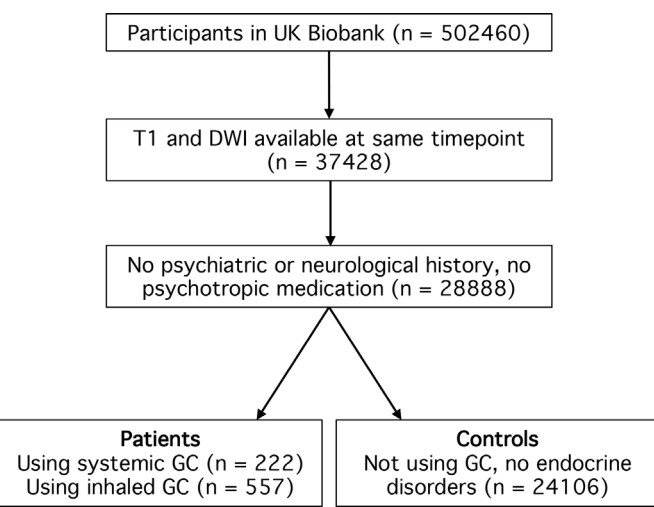

**Figure 1** Flow chart of participant inclusion. DWI, diffusion-weighted imaging; GC, glucocorticoids.

sagittal plane (voxel 1×1×1 mm; field-of-view 208×256×256 matrix). T1-weighted data were segmented using FAST (FMRIB's Automated Segmentation Tool),[42] to obtain volumes of cerebrospinal fluid, grey matter, and white matter, and to generate grey matter IDPs in 139 regions of interest (ROIs). Subcortical structures were modelled using FMRIB's Integrated Registration and Segmentation Tool (FIRST).[43] For this study, the mean volume of each bilateral structure was calculated over the two hemispheres, and the total cerebellar volume was calculated by adding up the volumes of all cerebellar lobules.

Diffusion imaging was performed using a standard Stejskal-Tanner pulse sequence to acquire 50 distinct diffusion-encoding directions for two diffusion-weighted shells with b values of 1000 and $2000 \, s/mm^2$ (voxel 2×2×2 mm; field-of-view 104×104×72 matrix). The $b=1000 \, s/mm^2$ data were fed into the diffusion-tensor-imaging (DTI) fitting tool (DTIFIT), which created DTI outputs including FA and MD. These outputs were then aligned to a standard-space white-matter skeleton using TBSS (Tract-Based Spatial Statistics)[44] and were averaged across a set of 48 standard-space tract masks defined by the John Hopkins University White Matter Atlas.[45] For this study, the mean FA and MD of each bilateral structure of interest were calculated over the two hemispheres. Moreover, global FA and MD measures were calculated by averaging these metrics over all white matter tracts per individual. Grey matter FA or MD were not available in the UK Biobank and are therefore not included in the global FA and MD.

### Cognitive and emotional data
At the assessment centres, participants also completed a series of cognitive tests and questionnaires on a touch screen. For these analyses, six cognitive tasks were selected: reaction time (to assess simple processing speed; expressed as mean time to correctly identify matches), trail making A and B (to test visual attention; expressed as the duration to complete the numeric (A)

or alphanumeric (B) path), fluid intelligence (to test reasoning and problem solving; expressed as a fluid intelligence score, which is the number of correct answers given to 13 questions), symbol digit substitution (to assess complex processing speed; expressed as the number of symbol digit matches made correctly within 2 min, with no maximum), and digit span (to test numeric working memory; expressed as the maximum number of digits remembered correctly, with a maximum of 12). For fluid intelligence, symbol digit substitution and digit span tests, higher scores represent a better cognitive performance, while for reaction time, and trail making A and B, higher scores represent a worse cognitive performance.

Moreover, we analysed four mental health questionnaire items that specifically asked about the participant's situation in the previous 2 weeks, in which the glucocorticoid users were likely already using glucocorticoid medication. These questions included the frequency of a depressed mood, disinterest, tenseness/restlessness, and tiredness/lethargy in the past 2 weeks, and were answered using categorical answer options ('Never', 'Several days', 'More than half of the days' or 'Nearly every day'). The entire questionnaire can be found via: https://biobank.ndph.ox.ac.uk/ukb/ukb/docs/TouchscreenQuestionsMainFinal.pdf.

### Statistical analysis
Demographic characteristics were presented as mean and SD or number and percentage and were compared across the three groups using analysis of variance (ANOVA) or chi squared tests, respectively.

The primary outcomes of this study were the differences in imaging parameters between glucocorticoid users and controls for a selection of ROIs (22 volumetric parameters, 14 diffusion parameters) that have previously been shown to be affected by long-term glucocorticoid exposure (see online supplemental file 2). As secondary outcomes, potential differences in cognitive and emotional outcomes between glucocorticoid users and controls were assessed.

The statistical analysis was performed in a stepwise approach, which is visualised in figure 2. For the imaging and cognitive outcomes, multivariable linear regression models were used. The assumption of normality of the residuals was assessed using quantile-quantile plots and homogeneity of variance across the groups was tested using Levene's test and was visually assessed using scatter plots. Subsequently, ANOVA was used to assess whether any differences in outcome parameters existed between systemic glucocorticoid users, inhaled glucocorticoid users and controls. To account for multiple testing, p values were adjusted using the Benjamini-Hochberg false discovery rate (FDR) method, for the number of comparisons tested (ie, 36 for imaging variables, 6 for cognitive variables). For those parameters with p values<0.05 after FDR correction, post hoc Dunnett tests were used to make pairwise comparisons between systemic glucocorticoid

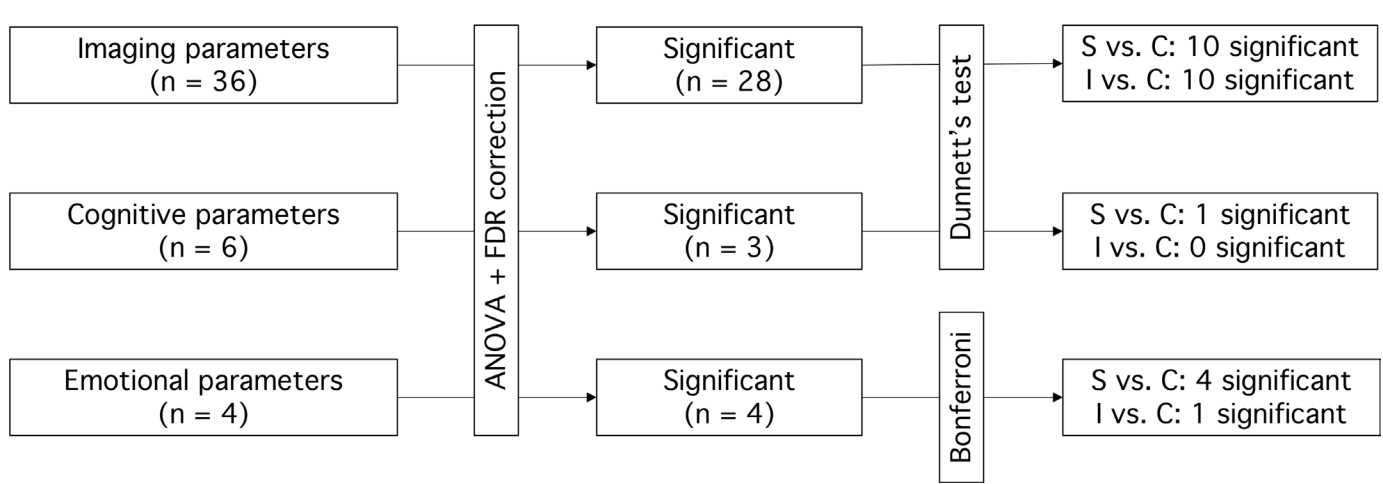

**Figure 2** Stepwise statistical analysis. ANOVA, analysis of variance; C, controls; FDR, false discovery rate correction; I, inhaled glucocorticoid users; n, number; S, systemic glucocorticoid users.

users versus controls, and inhaled glucocorticoid users versus controls.

For the multivariable linear models of the imaging parameters, covariates included age, sex, education, a measure of head size (the volumetric scaling from T1 image to standard space, corresponding to the inverse of head size), measures of head position (X position, Y position and Z position of the head in the scanner, and table position), assessment centre, and year of imaging acquisition. This selection was based on recommendations by the UK Biobank,[46] in addition to variables that potentially meet the criteria of a confounder for this study. Because fewer than 1% of the participants had missing values for the covariates, complete case analysis was performed for the analysis of these primary outcomes and all subsequent analyses. We considered that the very limited missing covariate data did not justify the intrinsic uncertainty that would come with imputation.

For the cognitive outcomes, variables with non-normally distributed residuals (reaction time, trail making A, trail making B) were normalised using log transformation. All cognitive outcomes were transformed such that higher values indicate better performance, and then converted into Z scores. The linear models of the cognitive outcomes were adjusted for age, sex and education.

Since the emotional outcome parameters were categorical, logistic models were used, adjusted for age, sex and education. Per symptom, the participants who reported a frequency of 'several days', 'more than half of the days' or 'nearly every day' were grouped together and were compared with participants who replied 'never'. The likelihood ratio test was performed to determine whether the proportion of patients experiencing a mental health complaint in the past 2 weeks differed between the three groups. For those parameters with a statistically significant difference after FDR correction (for four comparisons), the OR of experiencing a mental health complaint in the past 2 weeks was calculated for each glucocorticoid user group compared with controls. P values pertaining to the ORs were Bonferroni-corrected for multiple testing.

Use of glucocorticoids is associated with weight gain and in particular with an increased body fat percentage,[7] which has been reported to affect brain volume and white matter microstructure.[47] Therefore, mediation analysis was performed to test whether the association between glucocorticoid use and brain volume and white matter microstructure were mediated by body fat percentage (as measured by body impedance). For this analysis, all three significantly different volumetric outcomes, and the two (significantly different) global diffusion imaging parameters were considered. The mediation analysis was performed using the mediation package, with 1000 simulations and including the same covariates for the imaging parameters as above.

Since the doses and duration of medication use are unknown in the UK Biobank, we were unable to perform subgroup analyses based on dose or duration of glucocorticoid use. Because inhaled glucocorticoids are expected to cause, on average, lower systemic concentrations of glucocorticoids than orally or parenterally administered glucocorticoids,[48] the inhaled glucocorticoid users likely represent a group of patients exposed to lower systemic concentrations than patients using systemic glucocorticoids and might show less pronounced effects of glucocorticoids on brain parameters. This may give an indication of a dose-dependent effect of glucocorticoids on the brain. In addition, to assess whether we could identify potential duration-dependent or cumulative dose-dependent associations of glucocorticoid use with brain parameters, we performed an additional analysis in the subgroups of glucocorticoid users who reported using glucocorticoids at two different visits (before and including the imaging visit) and therefore likely represent a group of chronic or repeated glucocorticoid users. Since the low number of participants in this group expectedly resulted in a lower power, we performed the post hoc tests for these subgroups not only on those parameters that were statistically significant in the ANOVA, but on those parameters assessed by post-hoc tests in the main analysis, because this allowed us to gain insight into the difference in effect size compared with the main analysis.

van der Meulen M, *et al. BMJ Open* 2022;**12**:e062446. doi:10.1136/bmjopen-2022-062446

Lastly, to assess whether outlier values, possibly resulting from poor data quality or processing problems, affected the imaging or cognitive outcomes, the analyses were repeated while excluding outlier values of all outcome parameters (per outcome per study group), defined as more than 1.5 IQR below the first quartile or above the third quartile. For the cognitive parameters, the outliers were removed after transformation of the data. In addition, a sensitivity analysis of all outcome parameters was performed among all participants with imaging data available, without exclusion based on psychiatric, neurological, or endocrinological history, or medication use.

All statistical analyses and data visualisation were performed in R (V.4.1.1)[49] using the packages tidyverse (V.1.3.1),[50] car (V.3.0–11),[51] emmeans (V.1.7.0; https://cran.r-project.org/package=emmeans), lmtest (0.9–38),[52] mediation (V.4.5.0),[53] fauxnaif (V.0.6.1; https://cran.r-project.org/package=fauxnaif), ggpubr (V.0.4.0; https://rpkgs.datanovia.com/ggpubr/) and cowplot (V.1.1.1; https://cran.r-project.org/package=cowplot).

## Patient and public involvement

Patients and the public were not directly involved in the design or implementation of this study, since we used previously collected data.

## RESULTS

### Demographic characteristics

In total, 222 patients using systemic glucocorticoids, 557 patients using inhaled glucocorticoids, and 24106 controls were included. As shown in table 1, these groups did not differ significantly with respect to sex, education and smoking status, while the systemic glucocorticoid group was slightly older than the other groups (mean age 66.1±7.2 years for systemic glucocorticoid users; 63.3±7.5 years for inhaled glucocorticoid users; 63.5±7.5 years for controls), and the inhaled glucocorticoid group had a higher body mass index and body fat percentage (online supplemental file 3.1).

### Volumetric imaging parameters

Fifteen out of 22 predefined ROIs for the volumetric imaging were significantly different across the groups according to the ANOVA (online supplemental file 3.2). However, none of the 'global volume' parameters reached statistical significance in the post hoc tests (table 2, figure 3, online supplemental file 4). With respect to 'subcortical volumes', the caudate was larger in systemic glucocorticoid users compared with controls (adjusted mean difference (AMD)=77.8 mm$^3$, 95% CI 24.5 to 131.1). None of the subcortical volumes (containing both grey and white matter) differed significantly between inhaled glucocorticoid users and controls. Of the 'regional grey matter volumes', the caudate was larger in systemic glucocorticoid users compared with controls (AMD=178.7 mm$^3$, 95% CI 82.2 to 275.0), and inhaled glucocorticoid users had smaller grey matter volumes in the amygdala (AMD=−23.9 mm$^3$, 95% CI −41.5 to −6.2).

To assess whether chronic or repeated glucocorticoid exposure was associated with greater changes in imaging parameters, subgroup analyses among chronic systemic glucocorticoid users (n=42) and chronic inhaled glucocorticoid users (n=305) were performed (demographic characteristics are presented in online supplemental file 5). As expected, only few of the investigated imaging parameters reached statistical significance, potentially due to the lower power resulting from the smaller group sizes than in the main analysis (online supplemental file 3.3). Nevertheless, in chronic systemic glucocorticoid users, global volumes showed the same patterns of reduction as in the main analysis, and the caudate showed a larger increase in subcortical volume, but a smaller increase in grey matter volume. For chronic inhaled glucocorticoid users, the patterns were like those in the main analysis, with no striking differences in effect sizes (online supplemental file 3.3, online supplemental file 6 and 7).

### Diffusion imaging parameters

All but one of the diffusion imaging parameters differed significantly across the groups. Post hoc tests showed that systemic glucocorticoid use was associated with reduced global FA (AMD=−3.7e-3, 95% CI=−6.4e-3 to 1.0e-3), and reductions in regional FA were observed in the body and genu of the corpus callosum (table 2, figure 3, online supplemental file 4). Similarly, inhaled glucocorticoid use was associated with reduced global FA (AMD=−2.3e-3, 95% CI=−4.0e-3 to −5.7e-4), and the splenium of the corpus callosum and the cingulum of the hippocampus also showed a lower FA. For most ROIs, reductions in FA were smaller in inhaled glucocorticoid users than in systemic glucocorticoid users.

Furthermore, global MD was higher in systemic glucocorticoid users (AMD=7.2e-6, 95% CI=3.2e-6 to 1.1e-5) and inhaled glucocorticoid users compared with controls (AMD=2.7e-6, 95% CI=1.7e-7 to 5.2e-6). Systemic glucocorticoid was associated with higher regional MD in the body and genu of the corpus callosum, the cingulum of the hippocampus, and the uncinate gyrus. Inhaled glucocorticoid use showed significant associations with increased MD in the body, genu and splenium of the corpus callosum, the cingulum of the cingulate cortex, and the cingulum of the hippocampus. Again, effect sizes were similar or smaller for most tracts compared with the associations observed in systemic glucocorticoid users.

For chronic glucocorticoid users, the tendencies of FA and MD outcomes were in the same direction as the main analysis for all ROIs. Almost all associations with global and regional FA and MD showed a greater effect size among chronic systemic glucocorticoid users than in the main analysis, although only the global FA and MD measures, and FA and MD in the genu of the corpus callosum reached significance. In chronic inhaled glucocorticoid users, however, the effect sizes were not remarkably different from those observed in the main analysis

**Table 1** Characteristics of included patients using systemic glucocorticoids (n=222), inhaled glucocorticoids (n=557) and controls

| | Patients using systemic GC (n=222) | Patients using inhaled GC (n=557) | Controls (n=24 106) | P value (ANOVA) | Systemic GC vs controls* | | Inhaled GC vs controls* | |
|---|---|---|---|---|---|---|---|---|
| | | | | | Mean difference (95% CI) | P value | Mean difference (95% CI) | P value |
| Sex: male, n (%)† | 111 (50.0) | 253 (45.4) | 12 154 (50.4) | 0.066 | | | | |
| Age at time of scanning in years, mean (SD)† | 66.1 (7.2) | 63.3 (7.5) | 63.5 (7.5) | 2.4e-6 | 2.6 (1.4 to 3.7) | <0.0001 | −0.2 (−0.9 to 0.5) | 0.81 |
| Education level, n (%) | | | | 0.66 | | | | |
| College/university degree | 108 (48.6) | 287 (51.5) | 12 058 (50.0) | | | | | |
| A levels or equivalent | 26 (11.7) | 66 (11.8) | 2930 (12.2) | | | | | |
| O levels/GCSE or equivalent | 38 (17.1) | 96 (17.2) | 4155 (17.2) | | | | | |
| CSEs or equivalent | 9 (4.1) | 17 (3.1) | 879 (3.6) | | | | | |
| NVQ, HND, HNC or equivalent | 6 (2.7) | 35 (6.3) | 1396 (5.8) | | | | | |
| Other professional qualifications | 13 (5.9) | 29 (5.2) | 1150 (4.8) | | | | | |
| None of the above | 18 (8.1) | 25 (4.5) | 1311 (5.4) | | | | | |
| Missing | 4 (1.8) | 2 (0.4) | 227 (0.9) | | | | | |
| BMI in kg/m², mean (SD) | 26.2 (3.9) | 26.7 (4.3) | 26.1 (4.1) | 1.0e-3 | 0.0 (−0.6 to 0.7) | 0.98 | 0.6 (0.2 to 1.0) | 1.3e-3 |
| n (%) missing | 7 (3.2) | 20 (3.6) | 1325 (5.5) | | | | | |
| Body fat percentage, mean (SD) | 30.9 (7.9) | 32.1 (8.3) | 30.2 (7.9) | 3.5e-8 | 0.7 (−0.5 to 1.9) | 0.36 | 1.9 (1.1 to 2.7) | <1.0e-4 |
| n (%) missing | 7 (3.2) | 20 (3.6) | 1331 (5.5) | | | | | |
| Smoking status, n (%) | | | | 0.44 | | | | |
| Current | 6 (2.7) | 11 (2.0) | 647 (2.7) | | | | | |
| Previous | 76 (34.2) | 200 (35.9) | 7858 (32.6) | | | | | |
| Never | 137 (61.7) | 341 (61.2) | 15 380 (63.8) | | | | | |
| Missing | 3 (1.4) | 5 (0.9) | 221 (0.9) | | | | | |

*Calculated using post hoc Dunnett's test, only for those variables with a statistical difference according to the ANOVA.
†There were no missing values for the variables sex and age.
ANOVA, analysis of variance; BMI, body mass index; CSE, Certificate of Secondary Education; GC, glucocorticoids; GCSE, General Certificate of Secondary Education; HNC, Higher National Certificate; HND, Higher National Diploma; n, number; NVQ, National Vocational Qualification.

**Table 2** Imaging parameters, presented as the adjusted mean difference of patients using systemic glucocorticoids (GC) (n=222) or inhaled GC (n=557) compared with controls (n=24 106)

| | ANOVA | | | Systemic GC versus controls | | | Inhaled GC versus controls | | |
|---|---|---|---|---|---|---|---|---|---|
| | F value | P value | $P_{FDR}$ | AMD* | 95% CI | P value | AMD* | 95% CI | P value |
| **Volumetric measures** | | | | | | | | | |
| **Global volumes (in mm³)** | | | | | | | | | |
| Total brain volume | 19.7 | 2.8e-9 | **1.0e-8** | −3688 | −10627 to 3252 | 0.39 | 3374 | −1012 to 7760 | 0.16 |
| Grey matter volume | 23.7 | 5.4e-11 | **6.5e-10** | −1968 | −5904 to 1968 | 0.43 | 1012 | −1476 to 3500 | 0.57 |
| White matter volume | 6.7 | 1.2e-3 | **2.0e-3** | −1720 | −6273 to 2833 | 0.61 | 2362 | −516 to 5240 | 0.13 |
| Peripheral cortex | 21.1 | 6.9e-10 | **6.2e-9** | −3303 | −6843 to 237 | 0.072 | 1033 | −1205 to 3270 | 0.49 |
| CSF volume | 10.1 | 4.2e-5 | **9.5e-5** | 1215 | −824 to 3254 | 0.32 | 78 | −1211 to 1367 | 0.98 |
| **Subcortical volumes (in mm³)** | | | | | | | | | |
| Accumbens | 12.0 | 6.0e-6 | **1.7e-5** | −13.1 | −26.7 to 0.5 | 0.062 | −6.5 | −15.1 to 2.1 | 0.17 |
| Caudate | 6.7 | 1.3e-3 | **2.0e-3** | 77.8 | 24.5 to 131.1 | **0.0023** | −2.7 | −36.4 to 30.9 | 0.97 |
| Pallidum | 7.7 | 4.5e-4 | **7.8e-4** | 0.8 | −29.9 to 31.4 | 1.00 | −18.0 | −37.3 to 1.4 | 0.074 |
| Putamen | 10.9 | 1.8e-5 | **4.6e-5** | −31.3 | −98.2 to 35.6 | 0.48 | −27.9 | −70.2 to 14.4 | 0.25 |
| Thalamus | 8.2 | 2.7e-4 | **4.9e-4** | 3.6 | −74.0 to 81.1 | 0.99 | −6.4 | −55.4 to 42.6 | 0.93 |
| **Regional grey matter volumes (in mm³)** | | | | | | | | | |
| Amygdala | 23.8 | 5.0e-11 | **6.5e-10** | −4.0 | −31.9 to 23.8 | 0.91 | −23.9 | −41.5 to −6.2 | **5.2e-3** |
| Caudate | 13.0 | 2.3e-6 | **7.5e-6** | 178.7 | 82.2 to 275.0 | **1.0e-4** | 41.2 | −19.8 to 102.0 | 0.24 |
| Cerebellum | 10.8 | 2.0e-5 | **4.8e-5** | 25.1 | −18.4 to 68.5 | 0.34 | −12.2 | −39.7 to 15.3 | 0.51 |
| Insular cortex | 8.5 | 2.0e-4 | **3.9e-4** | −36.2 | −108.4 to 36.0 | 0.43 | 5.0 | −40.6 to 50.7 | 0.95 |
| Precuneal cortex | 5.5 | 4.3e-3 | **5.6e-3** | −21.5 | −179.0 to 136.3 | 0.92 | −7.4 | −107.0 to 92.4 | 0.97 |
| **DTI measures** | | | | | | | | | |
| **Fractional anisotropy** | | | | | | | | | |
| Global | 19.2 | 4.6e-9 | **2.8e-8** | −0.0037 | −0.0064 to −0.0010 | **4.2e-3** | −0.0023 | −0.0040 to −5.7e-4 | **5.7e-3** |
| Body of corpus callosum | 10.0 | 4.7e-5 | **1.0e-4** | −0.0043 | −0.0084 to −1.2e-4 | **0.043** | −0.0023 | −0.0049 to 3.0e-4 | 0.092 |
| Genu of corpus callosum | 16.8 | 5.4e-8 | **2.1e-7** | −0.0064 | −0.011 to −0.0017 | **5.0e-3** | −0.0019 | −0.0049 to 0.0011 | 0.27 |
| Splenium of corpus callosum | 5.4 | 4.4e-3 | **5.6e-3** | −0.0021 | −0.0053 to 0.0012 | 0.27 | −0.0032 | −0.0052 to −0.0012 | **1.0e-3** |
| Cingulum cingulate | 6.1 | 2.4e-3 | **3.4e-3** | −0.0017 | −0.0062 to 0.0028 | 0.61 | −0.0028 | −0.0057 to 8.9e-6 | 0.051 |
| Cingulum hippocampus | 6.4 | 1.7e-3 | **2.5e-3** | 6.5e-5 | −0.0046 to 0.0048 | 1.00 | −3.4e-3 | −0.0063 to −3.8e-4 | **0.024** |
| **Mean diffusivity** | | | | | | | | | |

Continued

**Table 2** Continued

| | ANOVA | | PFDR | Systemic GC versus controls | | | Inhaled GC versus controls | | |
|---|---|---|---|---|---|---|---|---|---|
| | F value | P value | | AMD* | 95% CI | P value | AMD* | 95% CI | P value |
| Global | 25.9 | 5.8e-12 | **2.1e-10** | 7.2e-6 | 3.2e-6 to 1.1e-5 | **1.0e-4** | 2.7e-6 | 1.7e-7 to 5.2e-6 | **0.034** |
| Body of corpus callosum | 15.5 | 2.0e-7 | **7.0e-7** | 6.9e-6 | 1.7e-6 to 1.2e-5 | **6.0e-3** | 4.8e-6 | 1.6e-6 to 8.1e-6 | **2.0e-3** |
| Genu of corpus callosum | 18.0 | 1.6e-8 | **7.0e-8** | 8.4e-6 | 2.2e-6 to 1.5e-5 | **4.9e-3** | 4.1e-6 | 1.7e-7 to 8.0e-6 | **0.039** |
| Splenium of corpus callosum | 9.7 | 6.2e-5 | **1.2e-4** | 4.4e-6 | −3.8e-8 to 8.9e-6 | 0.050 | 5.3e-6 | 2.4e-6 to 8.1e-6 | **1.0e-4** |
| Cingulum cingulate | 5.4 | 4.3e-3 | **5.6e-3** | 2.9e-6 | −8.5e-7 to 6.6e-6 | 0.16 | 2.8e-6 | 4.7e-7 to 5.2e-6 | **0.015** |
| Cingulum hippocampus | 18.5 | 9.1e-9 | **4.7e-8** | 5.0e-6 | 4.2e-7 to 9.5e-6 | **0.029** | 5.6e-6 | 2.8e-6 to 8.5e-6 | **<1.03e-4** |
| Uncinate fasciculus | 12.1 | 5.4e-6 | **1.6e-5** | 6.4e-6 | 2.2e-6 to 1.1e-5 | **1.4e-3** | 2.2e-6 | −4.4e-7 to 4.9e-6 | 0.12 |

P values in bold are statistically significant (p<0.05).
*Adjusted mean difference, calculated using linear models, adjusted for age, sex, education, X position, Y position and Z position of the head in the scanner, head size, assessment centre and year of imaging acquisition; significance was determined using a post hoc Dunnett's test.
AMD, adjusted mean difference; ANOVA, analysis of variance; CSF, cerebrospinal fluid; DTI, diffusion tensor imaging; $P_{FDR}$, Benjamini-Hochberg false discovery rate corrected p values.

(online supplemental file 3.3, online supplemental file 6 and 7).

### Cognitive and emotional outcomes

ANOVA showed differences between the groups on three cognitive tasks: trail making A, trail making B and symbol substitution (online supplemental file 3.4). Post hoc testing revealed that systemic glucocorticoid users performed significantly worse on the symbol digit substitution task compared with controls (AMD=−0.17 SD, 95% CI=−0.34 to −0.01; table 3). With regard to the emotional outcomes, between-group differences were observed in the frequency of depressive symptoms (p=0.0049), disinterest (p=0.0049), tenseness/restlessness (p=0.0025) and tiredness/lethargy (p=3.7e-7) (online supplemental file 3.5 and online supplemental file 8). Pairwise comparisons using logistic regression analysis revealed that systemic glucocorticoid users experienced more depressive symptoms (OR=1.76, 95% CI=1.25 to 2.43), disinterest (OR=1.84, 95% CI=1.29 to 2.56), tenseness/restlessness (OR=1.78, 95% CI=1.29 to 2.41), and tiredness/lethargy (OR=1.90, 95% CI=1.45 to 2.50) compared with controls (table 4), while inhaled glucocorticoid users only reported more tiredness/lethargy than controls (OR=1.35, 95% CI=1.14 to 1.60).

For the chronic users, none of the cognitive outcomes was significantly different in systemic or inhaled glucocorticoid users compared with controls in the post hoc analysis. Effect sizes for chronic systemic glucocorticoid users were even smaller than in the entire cohort, while two out of three were slightly larger in the chronic inhaled glucocorticoid users compared with the entire cohort (online supplemental file 3.6 and online supplemental file 9). Likewise, the emotional outcome parameters did not differ significantly, except for tiredness/lethargy which was more common in inhaled glucocorticoid users compared with controls. Remarkably, most ORs were lower than in the main analysis (online supplemental file 3.7 and online supplemental file 10).

### Sensitivity analyses

In the first sensitivity analysis we included the subjects that were previously excluded based on neurological, psychiatric or endocrine history or medication use. The imaging outcomes were comparable to those of the main analysis, with similar ROIs showing significant differences between the groups (online supplemental file 3.8–3.10, and online supplemental file 11–15), although the differences in diffusion parameters between glucocorticoid users and controls were more pronounced in the main analysis than in the unselected group. The same was observed for the cognitive and emotional outcomes.

For the second sensitivity analysis, outliers of the imaging and cognitive outcomes (<3% for most parameters) were excluded (online supplemental file 3.11–3.14 and online supplemental file 16 and 17), which led to the same conclusions for the imaging outcomes, except for a small number of regions that had shown a tendency in

van der Meulen M, *et al. BMJ Open* 2022;**12**:e062446. doi:10.1136/bmjopen-2022-062446

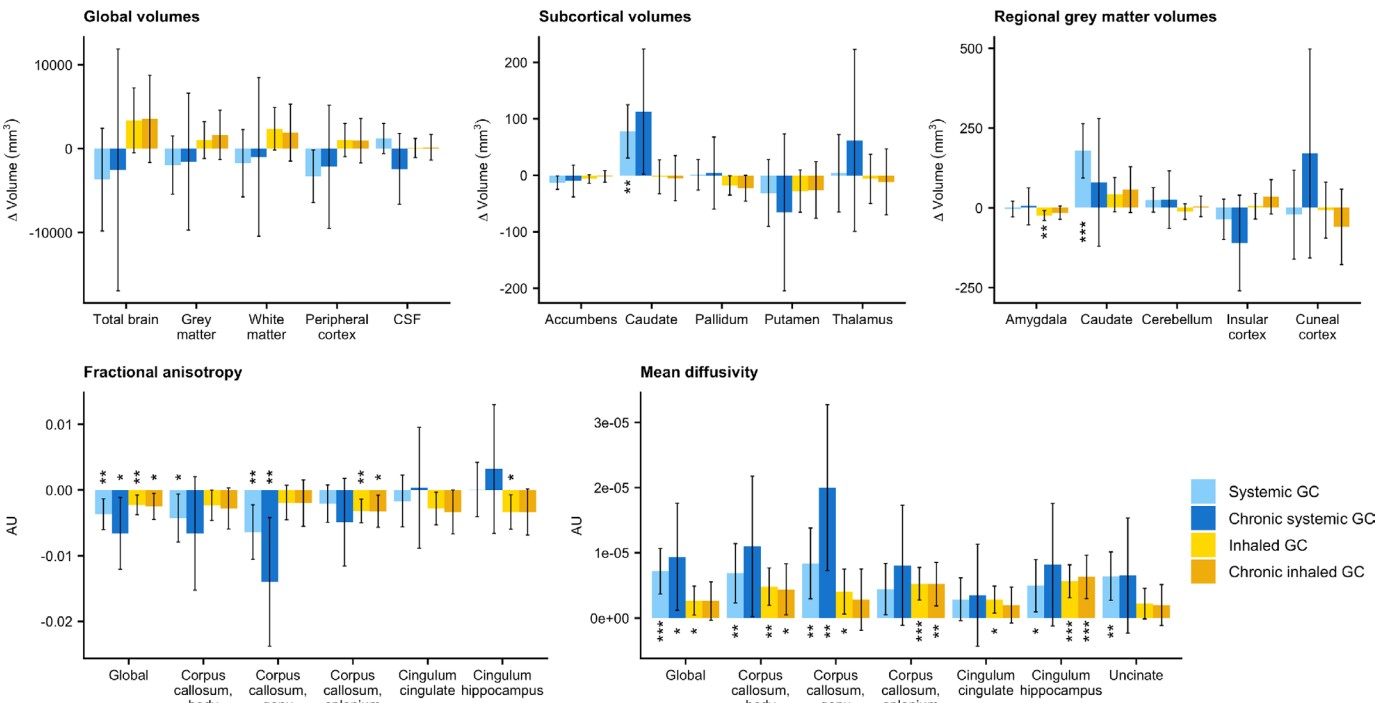

**Figure 3** Bar plots showing the adjusted mean difference (with 95% CI) of all imaging parameters for patients using systemic glucocorticoids (GC) (n=222) or inhaled GC (n=557), and subgroups of chronic systemic GC users (n=42), or chronic inhaled GC users (n=305) vs controls (n=24 106). Significance levels compared with controls: *p<0.05, **p<0.01, ***p<0.001. AU, arbitrary unit.

the main analysis and reached significance after exclusion of outlier values (subcortical accumbens volume, insular grey matter volume and MD in the splenium of the corpus callosum; all in systemic glucocorticoid users). For the cognitive outcomes, exclusion of outliers resulted in not only a significantly reduced score on the symbol digit substitution test, but also on the trail making B test for the systemic glucocorticoid users.

### Mediation analyses

To assess whether total body fat percentage could have mediated the association between glucocorticoid use and brain volume and white matter microstructure, mediation analysis was performed. For none of the investigated imaging outcomes was a significant mediation effect by body fat percentage found (online supplemental file 3.15), suggesting that the observed associations were independent of body fat.

### DISCUSSION

This study shows that in the large population-based cohort of the UK Biobank, the use of not only systemic glucocorticoids but also inhaled glucocorticoids is associated with changes in several brain imaging parameters. Most notably, the previously reported glucocorticoid effects on white matter microstructure[32] were also detected in this population and are therefore likely to be widespread among glucocorticoid users. Subgroup analyses among people using chronic glucocorticoids suggested a potential dose-dependent or duration-dependent effect

of glucocorticoids on white matter microstructure, with smallest effect sizes in inhaled glucocorticoid users, larger effect sizes in systemic glucocorticoid users, and the largest effect sizes in chronic systemic glucocorticoid users. While it remains unclear whether the observed effect sizes have clinical consequences for the population of glucocorticoid users as a whole, these findings are remarkable given the common neuropsychiatric side effects of synthetic glucocorticoids, and the observed changes may play a role in those patients suffering from these side effects.

### Findings in context

Previous studies in people exposed to high levels of endogenous glucocorticoids due to Cushing disease or high-dose synthetic systemic glucocorticoids have shown that glucocorticoid overexposure is associated with global cerebral atrophy and cortical thinning, as well as volumetric changes in specific brain areas. For example, reductions of grey matter volume have been observed in the hippocampus,[14 24 25 27 30 31 54] amygdala,[18 28 55] cingulate cortex,[13 22 23] insula,[13] caudate[19] and cerebellum,[17 25 26] which have all been implicated in cognitive processes and emotional regulation.[56–61] However, not all findings were consistent across studies, which may in part be due to differences between patient populations (eg, with respect to duration and type of glucocorticoid exposure), the small sample sizes of the studies, and the different analysis methods used, with some studies only focusing on one specific brain region, and others performing

**Table 3** Cognitive outcome measures of patients using systemic glucocorticoids (GC) (n=222) or inhaled GC (n=557) compared with controls

| | ANOVA | | | Systemic GC versus controls | | | Inhaled GC versus controls | | | Participants with available data, n (%) | | |
|---|---|---|---|---|---|---|---|---|---|---|---|---|
| | F value | P value | $P_{FDR}$ | AMD* | 95% CI | P value | AMD* | 95% CI | P value | Systemic GC | Inhaled GC | Controls |
| Trail making A | 5.6 | 0.0036 | **7.3e-3** | −0.11 | −0.28 to 0.06 | 0.25 | −0.031 | −0.15 to 0.09 | 0.78 | 149 (67) | 296 (53) | 16419 (68) |
| Trail making B | 6.1 | 0.0023 | **6.8e-3** | −0.12 | −0.30 to 0.05 | 0.19 | −0.0077 | −0.13 to 0.11 | 0.98 | 139 (63) | 291 (52) | 16071 (67) |
| Symbol substitution | 10.3 | 3.5e-5 | **2.1e-4** | −0.17 | −0.34 to −0.01 | **0.04** | −0.035 | −0.15 to 0.08 | 0.72 | 146 (66) | 298 (54) | 16442 (68) |

Trail making A, and trail making B were log transformed before generation of Z scores because they were non-normally distributed. Variables were transformed such that higher values indicate a better performance. Significance was determined using a post hoc Dunnett's test. P values in bold are statistically significant (p<0.05).
*Adjusted mean difference between patients and controls, expressed in Z scores. Calculated using linear models, adjusting for age, sex and education.
AMD, adjusted mean difference; ANOVA, analysis of variance; $P_{FDR}$, Benjamini-Hochberg false discovery rate corrected p values.

whole-brain analysis. In general, studies have mainly been dedicated to structural imaging with a specific interest in grey matter volume, while diffusion imaging has only been performed by a few studies in patients with Cushing disease.[32–35]

This study extends these findings by investigating brain volumes and white matter microstructure in not only systemic glucocorticoid users, but also inhaled glucocorticoid users, in whom neuropsychiatric side effects have been reported too.[62] The most remarkable and consistent associations were observed in white matter integrity, as both systemic and inhaled glucocorticoid use was associated with widespread reductions in FA and increases in MD. Although these associations were only about 10% of the effect sizes previously found in Cushing patients,[32] this adds to the growing body of literature suggesting that glucocorticoids have important impact on white matter, and that non-neuronal cells such as oligodendrocytes are very sensitive to glucocorticoids. Animal studies have shown that glucocorticoid exposure inhibits proliferation of oligodendrocyte progenitor cells throughout the white matter,[63] and induce changes in the expression of myelin basic protein, an oligodendrocyte marker.[36 64] Since oligodendrocytes are responsible for myelin production, glucocorticoid-induced changes in oligodendrocytes may underly the reduced white matter microstructure observed in patients using glucocorticoids. Besides oligodendrocytes, other glia cells including microglia and astrocytes are also affected by glucocorticoids, with multiple reports of decreased cell viability, proliferation, and immunoreactivity of microglia and astrocytes in response to glucocorticoids.[65–68]

Although we observed some patterns in global and regional brain volumes in glucocorticoid users, most of these did not reach significance. Rather surprisingly, although none of the global volumes was significantly different between patients and controls, the direction of change for all the areas was different for systemic (decreased volumes) vs inhaled glucocorticoid users (increased volumes). We did observe a significant association between inhaled glucocorticoid use and decreased grey matter volume of the amygdala, and systemic glucocorticoid use was associated with an increase in total and grey matter volume of the caudate nucleus. Decreased amygdala volumes have previously been reported in chronic systemic glucocorticoid users.[18 28 55] However, the increase in caudate volume contrasts with two previous studies that found larger caudate volumes after treatment of Cushing disease compared with during active disease,[19 20] while one other study reported an increased caudate volume in remitted patients compared with controls, but no differences in patients with active Cushing disease compared with controls.[21] Those findings suggest that cortisol excess caused a decreased caudate volume in these patients and/or that the caudate volume increased in response to normalisation of cortisol levels. The modest association of glucocorticoid use with brain volumes in the present population-based cohort

**Table 4** Likelihood of experiencing mental health complaints in the past 2 weeks of systemic glucocorticoid (GC) users (n=222) and inhaled GC users (n=557) compared with controls

| | Likelihood ratio test | | | Systemic GC vs controls | | | Inhaled GC vs controls | | |
|---|---|---|---|---|---|---|---|---|---|
| | $X^2$ | P value | $P_{FDR}$ | OR | 95% CI | P value | OR | 95% CI | P value |
| Depression | 10.6 | 0.0049 | 0.0049 | 1.76 | 1.25 to 2.43 | **8.2e-4** | 1.10 | 0.87 to 1.38 | 0.43 |
| Disinterest | 10.9 | 0.0043 | 0.0049 | 1.84 | 1.29 to 2.56 | **5.1e-4** | 1.06 | 0.82 to 1.36 | 0.64 |
| Tenseness | 13.4 | 0.0012 | 0.0025 | 1.78 | 1.29 to 2.41 | **3.0e-4** | 1.16 | 0.92 to 1.43 | 0.19 |
| Tiredness | 32.4 | 9.2e-8 | 3.7e-7 | 1.90 | 1.45 to 2.50 | **4.4e-6** | 1.35 | 1.14 to 1.60 | **6.3e-4** |

Calculated using logistic regression analysis, adjusting for age, sex, and education. P values in bold are statistically significant after Bonferroni correction for family-wise error rate of two tests (p<0.025).
$P_{FDR}$, Benjamini-Hochberg false discovery rate corrected p values.

study could indicate that white matter integrity is more sensitive to glucocorticoids than grey matter volume, and that longer or higher glucocorticoid exposure is needed to also induce volumetric changes.

It is tempting to relate these findings to glucocorticoid (GR) and mineralocorticoid receptor (MR) expression profiles in the brain. Previously, our group correlated the expression of GR and MR in several brain areas (data from the Allen Brain Atlas)[69] to the changes in brain volume observed in the extreme hypercortisolism caused by Cushing disease.[23] We then concluded that, although a high expression of these receptors was seen in the key brain areas such as the hippocampus, anterior cingulate cortex, and amygdala, there was no clear correlation between receptor expression profiles and brain areas affected by hypercortisolism. Receptor expression appears necessary but not predictive in this case. One might speculate that whether an area is affected by glucocorticoids may be more related to the densities of specific cell types that are responsive to glucocorticoids than the expression of receptors per se. Perhaps the density of oligodendrocytes, which are increasingly recognised as glucocorticoid-responsive, could be an important factor determining the responsiveness of different brain areas to glucocorticoids.

### Potential consequences and implications

It is well known that exogenous glucocorticoids are associated with neuropsychiatric side effects, including not only potentially severe mood disturbances such as depression and mania, but also cognitive impairment such as concentration and memory problems.[10] In this study, glucocorticoid users reported a higher frequency of several mental health complaints, while their cognitive performance was not significantly different, except for worse scores on the symbol digit substitution task in systemic glucocorticoid users. It should be noted that only a few mood-related parameters assessed by the UK Biobank were selected for this study, because these were the only parameters that applied specifically to the previous 2 weeks, in which the glucocorticoid users were likely already using their medication. Ideally, more aspects of mood would have been assessed to get a more comprehensive view

on the glucocorticoid users' psychological functioning. Furthermore, the observed mood-related effects may not be caused by glucocorticoid use per se but could also be related to the condition for which glucocorticoids were prescribed. For example, autoimmune and inflammatory diseases commonly treated with glucocorticoids, such as rheumatoid arthritis and chronic obstructive pulmonary disorder, have also been associated with mental health impairment and reduced quality of life.[70 71]

Nevertheless, awareness for the potential of glucocorticoids to affect the brain and cause neuropsychiatric symptoms is important, since these medications are prescribed for a wide range of conditions by many different medical specialties and are used by a substantial proportion of the population. Moreover, further research into the underlying mechanisms, reversibility, and risk factors for development of neuropsychiatric side effects of glucocorticoids is warranted, ideally considering dose and duration of glucocorticoids, as well as single-nucleotide polymorphisms (SNPs) in the GR gene (NR3C1) that affect glucocorticoid sensitivity. For those patients experiencing side effects, alternative treatment options should also be investigated. One promising direction is the development of selective GR modulators, since these (ideally) only activate the desired downstream signalling pathways in the desired cell types, limiting the potential side effects.[72 73]

### Strengths and limitations

To the best of our knowledge, this is the largest study to date assessing the association between glucocorticoid use and brain structure, and the first to investigate these associations in inhaled glucocorticoid users. For the selection of patients and controls, we applied relatively strict exclusion criteria to limit the potential confounding that may arise in observational cohort studies. Although not all neurological disorders, especially peripheral disorders, may have a clear impact on brain volume or white matter microstructure, UK Biobank participants with these conditions were excluded to prevent any confounding by these comorbidities. Our sensitivity analysis suggested that these conditions did not have a large impact on the results. However, we decided not to exclude patients with a history of depression, anxiety, mania or delirium,

 

because these are known possible consequences of glucocorticoid use,[10] and we did not want to exclude patients based on potentially glucocorticoid-related outcomes.

Another method used to limit confounding was adjustment of the regression analyses for relevant confounding variables, including demographic variables and variables related to the imaging visits (eg, assessment centre, position of the head in the scanner). For both the volumetric and diffusion parameters, head size was used as covariate, because previous research not only found a relation between head size and brain volume, but also between head size and DTI parameters.[74 75] The use of this variable as covariate is also recommended by the UK Biobank.[46] We decided not to include a measure of body weight or body composition as covariate, because it is known that glucocorticoids can cause obesity,[7] which is therefore more likely to be in the causal pathway than to be a confounder. Our mediation analysis, however, suggested that body fat percentage did not mediate the associations identified. Nevertheless, despite the correction for a wide range of potential confounders, it should be noted that the possibility of residual confounding cannot be excluded.

In addition, although a causal relation between glucocorticoid use and changes in the brain is likely based on the present and previous studies, the cross-sectional nature of this study does not allow for formal conclusions on causality. Demonstrating a dose–response effect of glucocorticoid on imaging parameters would have increased the likelihood of a causal relation, but unfortunately, dose and duration of medication use were not available in the UK Biobank. We were therefore only able to give an indication of a dose–response effect by performing separate analyses in systemic glucocorticoid users, inhaled glucocorticoid users (representing a group exposed to lower systemic concentration of glucocorticoids), and subgroups of patients using systemic or inhaled glucocorticoid chronically (representing groups with a longer duration and larger cumulative dose of glucocorticoid use). The fact that the effect sizes of the associations between glucocorticoid use and diffusion imaging parameters are generally largest in the chronic systemic glucocorticoid group, and smallest in the inhaled glucocorticoid group, indicates that a dose-dependent or duration-dependent effect may exist, although the expected lower power of the small chronic systemic glucocorticoid group likely precluded most associations from reaching significance. Moreover, while the association effect size estimates were larger in chronic systemic glucocorticoid users compared with the main group using systemic glucocorticoids, this difference was not observed among inhaled glucocorticoid users. A potential explanation may be that inhaled glucocorticoids are generally prescribed for a longer duration than systemic glucocorticoids, which is also reflected by the high percentage of inhaled glucocorticoid users (326/592, 55%) that could be included in the subgroup of chronic users, compared with the lower percentage of chronic systemic glucocorticoid users (48/234, 21%).

Another limitation is that we could not differentiate between oral and parenteral glucocorticoids because of the medication names used by the UK Biobank. We were, therefore, unable to conduct separate analyses for these groups and analysed them together as systemic glucocorticoid users. Also, 14 participants used both inhaled and systemic glucocorticoids. Since this group was too small to analyse separately in a meaningful way, these participants were included in the systemic group. Although simultaneous use of different glucocorticoids might be associated with more profound changes in the brain, we do not expect that this association is larger than the effect size differences that may exist because of differences in dosages of the systemic glucocorticoids. Lastly, some seasonal patterns in glucocorticoid use may exist depending on the indications, which we were unable to adjust for in the analyses.

## CONCLUSION

This study shows that both systemic and inhaled glucocorticoids are associated with an apparently widespread reduction in white matter integrity, which may in part underly the neuropsychiatric side effects observed in patients using glucocorticoids. Since these medications are widely used, awareness of these associations is necessary across medical specialties and research into alternative treatment options is warranted.

**Acknowledgements** We would like to thank Dr. Roula Tsonaka for her advice regarding the statistical analysis, and Dr. Steven van der Werff for his suggestion for this study.

**Contributors** JMA and OCM contributed to the study conception. MvdM designed the study, performed the analyses, and wrote the first version of the manuscript. OMD contributed to the statistical analyses. All authors read and commented on the manuscript and approved the final version of the manuscript. MvdM and OCM are the guarantors of the manuscript and accept full responsibility for the work and conduct of the study, had access to the data, and controlled the decision to publish. The corresponding author attests that all listed authors meet authorship criteria and that no others meeting the criteria have been omitted.

**Funding** The UK Biobank was established by the Wellcome Trust, Medical Research Council, Department of Health, Scottish government, and Northwest Regional Development Agency. It also received funding from the Welsh Government, the British Heart Foundation, Cancer Research UK, and Diabetes UK. For the analyses presented in this manuscript, MvdM received a personal MD/PhD grant of the Leiden University Medical Center, and JMA received support from CONACyT (the National Council for Science and Technology-Government of Mexico).

**Disclaimer** The funding sources had no role in the study conduct, data collection, analyses, data interpretation and the decision to submit the manuscript.

**Competing interests** All authors have completed the ICMJE uniform disclosure form at http://www.icmje.org/disclosure-of-interest/ and declare: MvdM received financial support from the MD/PhD grant of the Leiden University Medical Center, and JMA received financial support from CONACyT (the National Council for Science and Technology-Government of Mexico) for the submitted work; OCM has received research grants and honorariums from Corcept Therapeutics, and a speakers' fee from Ipsen; no other relationships or activities that could appear to have influenced the submitted work.

**Patient and public involvement** Patients and/or the public were not involved in the design, or conduct, or reporting, or dissemination plans of this research.

**Patient consent for publication** Not applicable.

**Ethics approval** This study was performed under the ethical approval obtained by UK Biobank from the National Health Service National Research Ethics Service (ref

11/NW/0382, 17 June 2011). Data for the present study were obtained from the UK Biobank under application number 59004. Participants gave informed consent to participate in the study before taking part.

**Provenance and peer review** Not commissioned; externally peer reviewed.

**Data availability statement** Data may be obtained from a third party and are not publicly available. Data used for this study are available via application to the UK Biobank.

**ORCID iD**
Merel van der Meulen http://orcid.org/0000-0002-0001-4408

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
