## [Reviewer comments · BMJ Open]

ARTICLE DETAILS

TITLE (PROVISIONAL)	Association between use of systemic and inhaled glucocorticoids and changes in brain volume and white matter microstructure: a cross-sectional study using data from the UK Biobank
AUTHORS	van der Meulen, Merel; Amaya, Jorge Miguel; Dekkers, Olaf; Meijer, Onno C.

VERSION 1 – REVIEW

REVIEWER	Goldwaser, Eric University of Maryland Baltimore, Psychiatry
REVIEW RETURNED	20-Apr-2022

GENERAL COMMENTS	It is my pleasure to review the original research article, "Oral and inhalation glucocorticoid use associate with changes in brain volume and white matter microstructure: a cross-sectional UK Biobank study" by van der Meulen and colleagues. Authors utilized the well-established UK Biobank for the purposes of collecting cross-sectional data in large cohorts of glucocorticoid users and controls to assess for differences in various DTI neuroimaging correlates of interest. They secondarily analyzed data in subgroups for oral vs inhalation glucocorticoid users as well as with cognitions and symptoms between groups for structural neuroimaging measures. Main findings include an association of glucocorticoid use with reductions in white matter FA in both cohorts (oral and inhalation), as well as increased MD vs controls. Regional analyses demonstrated increased caudate size with oral glucocorticoid use and reduced volume of the amygdala in inhalation glucocorticoid users compared to controls. Oral users moreover demonstrated poorer performance on processing speed and worsened depressive symptoms, disinterest, tenseness/restlessness, and tiredness/lethargy vs controls. Strengths identified by the authors included the large dataset and narrow inclusion criteria, and a notable weakness was that dose and duration of medication was not known. The main emphasis was on broadening and generalizing findings from smaller studies that linked glucocorticoid use to brain structural abnormalities. This is a well-written paper, with an obvious importance in the research question. High level of rigor was performed in their data analyses. There is enthusiasm for this paper and the findings, which have helped move the general knowledgebase forward in significant ways. Several key details are lacking that should be addressed prior to accepting the authors' conclusions, however. Notably, more detailed graphical representations of the data would be helpful in interpreting associations, which the figures/results could benefit from. There is a wealth of exploratory data, and some of the statistical parameters used to minimize type I error will need further clarity. There also appears a discrepancy in the exclusion criteria. If the major
--

comments below are accounted for, I believe this paper will have wide-reaching attention and may serve important follow up, hypothesis-driven work.

Introduction

-could benefit from further background information about how FA and MD are interpreted and what previous findings have shown, to support the hypothesis developed and tested here, in addition to the structural volumetric background.

-how do FA and MD relate, compare, and contrast to one another – it would be helpful, especially given the audience is not necessarily expected to know how to interpret FA and MD values, what previous literature has found on these DTI measures and GC use/cognitions/symptoms, and what the expected outcomes were for the hypothesis.

-no information on injection glucocorticoids is given, which is curious why these are left out. It may serve as an interesting 'intermediate' group, as injection glucocorticoids are typically much shorter duration of use (on the order of several days or a few weeks) and associated with psychiatric presentations in acute settings.

-overall, the intro lacks a more fully developed background to set up the hypothesis being tested, and what are the expected outcomes, including directionality, of the measures.

Methods

-under data collection, was missing data imputed? How was it accounted for in scales/neuropsych testing?

-selection criteria seems somewhat arbitrary for exclusion of psychiatric disease if the reason stated is because "may be related to glucocorticoid use". This is very vague and not explicit. Psychosis was not included in the list? Also neurologic conditions were exclusionary. What about chronic infections (HIV, hepatitis) or substance use disorder? There is a much more extensive list of conditions that are shown in the exclusion criteria in the supplemental..but it is not listed here, which needs to be clarified.

Are injection glucocorticoids included in the control group? Were use of injection glucocorticoids accounted for in either group? Use of oral or inhalation glucocorticoids were criteria, however not use of both? It is likely that within this large cohort there were users of both oral and inhalation glucocorticoids..how were they accounted for?

-global FA and MD was calculated by "averaging over all white matter tracts" – does this mean that gray matter FA and MD was not included in these whole-brain values?

-FDR correction for how many comparisons?

-Not clear why BF correction is used at times (in the secondary analyses), and FDR correction is used at other times (primary outcomes) for multiple comparisons. Please clarify or perhaps consider sticking to one method, and be more explicit as to how many comparisons are used in the calculation each time it is applied.

-Why is education included as covariate with age and sex for all analyses? It seems rather exploratory and non-standard, especially if group differences were not observed. For better generalizability of findings, it would be suggested to use age and sex, and include education in the model as a separate analysis where appropriate (for cognitive/neuropsych for example).

-use of inhalation glucocorticoids may have a much more seasonal pattern of use given the indications, and as such, there may be a heavy bias related to when data was collected temporally. The two-week timeframe may not truly capture chronicity of use if inhalation glucocorticoids is used for say, seasonal allergies, or temperature-

induced asthmatic reactions. Was this considered or gathered by the authors? It may be an important limitation to include otherwise.
-authors mention that by including a more representative group of chronic users they had constrained their power to detect changes. Was power analysis performed to validate this claim?

Results

-given the primary outcome measure is glucocorticoid users (oral and inhalation inclusive) vs controls, it would make sense to have the first table formatted to show demographics and clinical descriptive statistics of these two group comparisons. A separate table or format would be helpful to then show what the oral vs inhalation subgroups are comprised of for the secondary/exploratory analyses.

-it is stated that "...only few of the investigated imaging parameters reached statistical significance due to the lower power resulting from...". Is it known that the non-significance is due to reduced power? Unless further statistical testing is done to show this, it may not be an appropriate conjecture to make.

-table 2: should standardize format of scientific notation throughout. Also, explicitly state how many comparisons are included in the FDR correction. Based on the table, it seems FDR correction was only applied to the ANOVA comparison, while post-hoc testing of oral GC vs controls used uncorrected P-values? If so, it is not likely that $p=0.002$ would survive significance correction for the caudate..this will need clarification given it is highlighted throughout the paper if I understand the analysis correctly.

-it is unclear why education is included in the adjusted means. I think it would be helpful to consider removing it if it does not add any scientific value to the model.

-clarification is needed on the exclusion of outliers for the cognitive tests before the findings can be validly interpreted: were the outliers from the non-transformed data? Were outliers excluded iteratively for each measure tested or independently for each statistical test performed? Can sample size be included in the text for completeness?

-are the cognitive/neuropsych tests standardized/scaled scores that adjust for age? Or is it raw scores used?

-figure 2 has extremely small font in the bar graphs, making it difficult to read. Significance levels for group comparisons should be included in the bar chart.

-graphical representations of numerical data, especially neuroimaging findings, would be advised to present in the form of a brain template if able, to better understand the neuroanatomic distribution of the findings. At the very least, scatter plots would be helpful to interpret the patterns in key findings.

Discussion

-expression and concentration of GRs in the brain have region specificity. Is there a way to further categorize the findings, even the counter-intuitive ones like increased caudate volume, with receptor expression profiles? Some discussion about the differential patterns of GR throughout key brain regions, like hippocampus/amygdala, PFC, etc., may be warranted to further illustrate and explain the results.

-in the strengths/limitations portion, it is stated that psychiatric diseases like anxiety, depression, etc., were not excluded..but in the participants section of the method it states that they were excluded. It is a bit confusing to me how it is worded. Although it appears the rationale made explicit in the discussion states that the authors did

	not want to exclude patients based on potentially glucocorticoid-related outcomes, implying that these patients were not excluded, as they state here. Consider simplifying the language used in the methods section if that is the case..again, psychosis/schizophrenia is not deliberately stated amongst these conditions, which is curious and should be addressed as to why, however is included in the conditions listed in the supplemental. -in discussing potential confounds, it would be worthwhile to mention about the use of education as a covariate in analyses. Most of the cognitive/neuropsych testing has standardized scaled scores that do adjust for age already..but it's not clear if this is already done or the results are raw scores. Level of education may be a valid covariate in these analyses, but it is not known to me how conventional that is. May be helpful to note if it is going to be used in this specific analysis as well as in the analysis of the DTI and glucocorticoid relationships with and without education level as a covariate. -the note that the limited sample size did not allow for significant findings in the chronic oral glucocorticoid group may be in error. Reference to other studies better powered for this specific claim may lend credence to it, however power alone was not assessed to make this determination. Authors can consider performing a power analysis to make such an inference in this subgroup. Conclusion -the association noted as a main conclusion is overly broad – ‘decreased white matter integrity’ should be more precisely defined based on region specificity the authors found to be most conclusive and important.
--	--

REVIEWER	Kalafatakis, Konstantinos Panepistimio Ioanninon
REVIEW RETURNED	02-May-2022

GENERAL COMMENTS	This is a very interesting research effort, that utilizes UK Biobank data to study the effects of corticosteroid treatment (oral & inhaled) on MRI and behavioural/ psychological markers of brain structure/ function. This must be indeed the largest study to date assessing the effects of glucocorticoid on the brain, especially when it comes to inhaled steroids (which are very frequently used in COPD). My only remark would be the following: I think the readers would benefit a lot by adding another Figure which illustrates step-by-step the (MRI and rest of) data processing pipeline, that was followed in each case. That way, the readers would see 3 Figures, one related to subject inclusion and numbers (already there), one on data curation, processing and analysis pipelines, and on with the results (already there).
--

VERSION 1 – AUTHOR RESPONSE

Reviewer: 1

Dr. Eric Goldwaser, University of Maryland Baltimore

Comments to the Author:

It is my pleasure to review the original research article, “Oral and inhalation glucocorticoid use associate with changes in brain volume and white matter microstructure: a cross-sectional UK Biobank study” by van der Meulen and colleagues. Authors utilized the well-established UK Biobank for the purposes of collecting cross-sectional data in large cohorts of glucocorticoid users and controls to assess for differences in various DTI neuroimaging correlates of interest. They

secondarily analyzed data in subgroups for oral vs inhalation glucocorticoid users as well as with cognitions and symptoms between groups for structural neuroimaging measures. Main findings include an association of glucocorticoid use with reductions in white matter FA in both cohorts (oral and inhalation), as well as increased MD vs controls. Regional analyses demonstrated increased caudate size with oral glucocorticoid use and reduced volume of the amygdala in inhalation glucocorticoid users compared to controls. Oral users moreover demonstrated poorer performance on processing speed and worsened depressive symptoms, disinterest, tenseness/restlessness, and tiredness/lethargy vs controls. Strengths identified by the authors included the large dataset and narrow inclusion criteria, and a notable weakness was that dose and duration of medication was not known. The main emphasis was on broadening and generalizing findings from smaller studies that linked glucocorticoid use to brain structural abnormalities. This is a well-written paper, with an obvious importance in the research question. High level of rigor was performed in their data analyses. There is enthusiasm for this paper and the findings, which have helped move the general knowledgebase forward in significant ways. Several key details are lacking that should be addressed prior to accepting the authors' conclusions, however. Notably, more detailed graphical representations of the data would be helpful in interpreting associations, which the figures/results could benefit from. There is a wealth of exploratory data, and some of the statistical parameters used to minimize type I error will need further clarity. There also appears a discrepancy in the exclusion criteria. If the major comments below are accounted for, I believe this paper will have wide-reaching attention and may serve important follow up, hypothesis-driven work.

Introduction

1. Could benefit from further background information about how FA and MD are interpreted and what previous findings have shown, to support the hypothesis developed and tested here, in addition to the structural volumetric background.
2. How do FA and MD relate, compare, and contrast to one another – it would be helpful, especially given the audience is not necessarily expected to know how to interpret FA and MD values, what previous literature has found on these DTI measures and GC use/cognitions/symptoms, and what the expected outcomes were for the hypothesis.

To address these two remarks, we have added the following section to the Introduction (p.4, lines 109-113):

“In humans, this was studied using diffusion tensor imaging (DTI), showing globally decreased fractional anisotropy (FA), which represents the directionality of water diffusion through the brain and is a marker of microstructural architecture³⁷, and increased mean diffusivity (MD)³²⁻³⁵, which represents an increase in water diffusion in all directions and is associated with disease processes such as inflammation and oedema³⁷.”

To specify our hypotheses, we also added the following to the Introduction (p.4, lines 122-125):

“Based on previous literature, we hypothesized that glucocorticoid use would be associated with decreased grey matter volumes in the limbic system and hippocampus, a widespread reduction in FA and increase in MD throughout the brain, and poorer cognitive and emotional outcomes.”

3. No information on injection glucocorticoids is given, which is curious why these are left out. It may serve as an interesting 'intermediate' group, as injection glucocorticoids are typically much shorter duration of use (on the order of several days or a few weeks) and associated with psychiatric presentations in acute settings.

We agree with the reviewer that this would be a valuable third comparison. However, unfortunately, in the UK Biobank no separate injection glucocorticoid medications are coded. As can be seen in Supplement 1, some medication codes in the UK Biobank are very general, without clarification of route of administration for specific drugs that can be administered both orally and parenterally. It is therefore likely that some of the people included in the 'oral' group received the medication via

injection or infusion. We therefore decided to change the name of this group to 'systematic glucocorticoid users' and have replaced 'oral' by 'systemic' throughout the manuscript. Moreover, we clarified in the Methods section that the 'systemic users' include both oral and parenteral glucocorticoids (p. 5, lines 162-165):

"Individuals who met these criteria and used oral or parenteral glucocorticoids at the time of imaging were included in the systemic glucocorticoid patient group (n = 222), and individuals who met these criteria and used inhalation glucocorticoids (but no systemic glucocorticoids) at the time of imaging were included in the inhalation glucocorticoid group (n = 557)."

We also added this as a limitation in the Discussion (p. 25, lines 856-859):

"Another limitation is that we could not differentiate between oral and parenteral glucocorticoids because of the medication names used by the UK Biobank. We were therefore unable to conduct separate analyses for these groups and analysed them together as systemic glucocorticoids."

4. Overall, the intro lacks a more fully developed background to set up the hypothesis being tested, and what are the expected outcomes, including directionality, of the measures.

We hope to have sufficiently addressed this question in our reply to comments 1 and 2 above.

Methods

5. Under data collection, was missing data imputed? How was it accounted for in scales/neuropsych testing?

Because we only selected patients with available imaging data, there were no missing data for the primary outcome variables. As stated in the Statistical analysis section (p.8, lines ...), since fewer than 1% of the participants had missing values for the covariates, complete case analysis was performed for the analysis of the primary outcomes and all subsequent analyses. We considered imputation but decided against it, since imputation itself introduces a degree of uncertainty. Especially with such low percentages of missing data, we felt that the added value of a slightly more data points would not weigh up against the additional inaccuracy. Similarly, for the secondary outcomes complete case analysis was used.

We clarified this in the Methods section (p.8, lines 260-262):

"We considered that the very limited missing covariate data did not justify the intrinsic uncertainty that would come with imputation."

6. Selection criteria seems somewhat arbitrary for exclusion of psychiatric disease if the reason stated is because "may be related to glucocorticoid use". This is very vague and not explicit. Psychosis was not included in the list? Also neurologic conditions were exclusionary. What about chronic infections (HIV, hepatitis) or substance use disorder? There is a much more extensive list of conditions that are shown in the exclusion criteria in the supplemental..but it is not listed here, which needs to be clarified. Are injection glucocorticoids included in the control group? Were use of injection glucocorticoids accounted for in either group? Use of oral or inhalation glucocorticoids were criteria, however not use of both? It is likely that within this large cohort there were users of both oral and inhalation glucocorticoids..how were they accounted for?

In general, it can be assumed that any brain disease can be a confounder for brain imaging. Therefore, for the main analysis, we excluded as many brain-related, psychiatric, and

neurological diseases as possible. However, four psychiatric conditions are most commonly described as potential consequences of glucocorticoid use, i.e., anxiety, depression, mania, and delirium (Judd et al., 2014). For that reason, we did include patients who suffered from these psychiatric conditions.

We clarified our reason to do so in the manuscript (p.5, lines 156-159):

“We selected participants who [...] did not have a history of psychiatric disease based on self-reported data or HES data. However, we did include the psychiatric diseases most commonly associated with glucocorticoid use based on previous literature (anxiety, depression, mania, and delirium) ¹⁰ as we did not want to exclude patients based on potentially glucocorticoid-related outcomes”

Moreover, to test whether the outcomes would be different if all patients regardless of psychiatric/neurological history were included, we performed a sensitivity analysis as described in the Statistical analysis section (p.9, lines 306-309):

“In addition, a sensitivity analysis of all outcome parameters was performed among all participants with imaging data available, without exclusion based on psychiatric, neurological, or endocrinological history, or medication use.”

We also modified the Methods section to clarify that in the inhaled glucocorticoids group, only inhaled glucocorticoids were allowed, while the few people using both inhaled and systemic glucocorticoids were included in the systemic glucocorticoids group, since we expected that the systemic medication would have larger impact on brain parameters. This group was too small to include as a separate group (p.5, lines 162-166):

“Individuals who met these criteria and used oral or parenteral glucocorticoids at the time of imaging were included in the systemic glucocorticoid patient group (n = 222), and individuals who met these criteria and used inhaled glucocorticoids (but no systemic glucocorticoids) at the time of imaging were included in the inhaled glucocorticoid group (n = 557). Among the patients using systemic glucocorticoids, 14 were also using inhaled glucocorticoids.”

We have also added this as a limitation to the Discussion (p.25, lines 859-863):

“Also, 14 participants used both inhalation and systemic glucocorticoids. Since this group was too small to analyse separately in a meaningful way, these participants were included in the systemic group. Although simultaneous use of different glucocorticoids might be associated with more profound changes in the brain, we do not expect that this effect is larger than the effect size differences that may exist because of differences in dosages of the systemic glucocorticoids.”

7. Global FA and MD was calculated by “averaging over all white matter tracts” – does this mean that gray matter FA and MD was not included in these whole-brain values?

Indeed, the FA and MD in the UK Biobank were only measured over white matter tracts, so no grey matter FA and MD were analysed. We have specified this in the Methods section (p.6, lines 205-206):

“Grey matter FA or MD were not available in the UK Biobank and are therefore not included in the global FA and MD.”

8. FDR correction for how many comparisons?

FDR correction was performed for the number of comparisons tested for each group, i.e., 36 for imaging variables, 6 for cognitive variables, and 4 for emotional variables. We have specified this in the manuscript:

“To account for multiple testing, P values were adjusted using the Benjamini-Hochberg false discovery rate (FDR) method, for the number of comparisons tested (i.e., 36 for imaging variables, 6 for cognitive variables).” (p.7, lines 246-248)

“For those parameters with a statistically significant difference after FDR correction (for 4 comparisons)...” (p.8, lines 272-273)

9. Not clear why BF correction is used at times (in the secondary analyses), and FDR correction is used at other times (primary outcomes) for multiple comparisons. Please clarify or perhaps consider sticking to one method, and be more explicit as to how many comparisons are used in the calculation each time it is applied.

We used a stepwise approach in our statistical analysis. First, as described in the Statistical analysis section, for both primary and secondary analyses FDR correction was used. Subsequently, for those variables that were significant after FDR correction, a post-hoc analysis was performed. For the continuous parameters (imaging and emotional outcomes), this was done using Dunnett’s test. For the emotional parameters, that are expressed as odds ratios, Dunnett’s test was not possible, and post-hoc analysis was performed using Bonferroni correction for groupwise comparisons. To clarify this stepwise approach, we have added another figure (new Figure 2) in which the statistical analysis is visualized.

10. Why is education included as covariate with age and sex for all analyses? It seems rather exploratory and non-standard, especially if group differences were not observed. For better generalizability of findings, it would be suggested to use age and sex, and include education in the model as a separate analysis where appropriate (for cognitive/neuropsych for example).

We have included education as a covariate because education is related to intelligence and cognitive functioning. We feel that the scientific value of adding this covariate to the model lies in the assumption that every difference in brain function (i.e., intelligence, cognition) potentially has a brain substrate and therefore should be accounted for in the analyses.

11. Use of inhalation glucocorticoids may have a much more seasonal pattern of use given the indications, and as such, there may be a heavy bias related to when data was collected temporally. The two-week timeframe may not truly capture chronicity of use if inhalation glucocorticoids is used for say, seasonal allergies, or temperature-induced asthmatic reactions. Was this considered or gathered by the authors? It may be an important limitation to include otherwise.

We thank the reviewer for this insight, which we had not considered before. Unfortunately, it is not possible to correct for this in this dataset. We have therefore included this as a limitation in the Discussion (p.25, lines 863-865):

“Lastly, some seasonal patterns in glucocorticoid use may exist depending on the indications, which we were unable to adjust for in the analyses.”

12. Authors mention that by including a more representative group of chronic users they had constrained their power to detect changes. Was power analysis performed to validate this claim?

Power analysis requires a realistic estimate of the effect size, which was impossible to define. If we had used the effect size observed in the total cohort as estimate for the chronic users, we could have expected this to be an underestimation – as was confirmed in the analysis of the chronic users. Importantly, power analysis is generally performed a priori to define the required sample size. This situation does not apply to our study using UK Biobank data, since we were not able to adjust the number of included participants. Our statement that by analysing a subgroup of our total cohort, we constrained the power to detect changes, follows logically from the power calculation formula: the fewer participants, the lower the power. We modified the wording in the Statistical analysis section slightly to nuance our statement (p.9, lines 297-298):

“Since the low number of participants in this group expectedly resulted in a lower power...”

Results

13. Given the primary outcome measure is glucocorticoid users (oral and inhalation inclusive) vs controls, it would make sense to have the first table formatted to show demographics and clinical descriptive statistics of these two group comparisons. A separate table or format would be helpful to then show what the oral vs inhalation subgroups are comprised of for the secondary/exploratory analyses.

We agree with the reviewer that this would be insightful. We had already presented these comparisons in Supplement 3.1 but have now also added the data to Table 1.

14. It is stated that “...only few of the investigated imaging parameters reached statistical significance due to the lower power resulting from...”. Is it known that the non-significance is due to reduced power? Unless further statistical testing is done to show this, it may not be an appropriate conjecture to make.

We agree with the reviewer that this should be stated less firmly and have therefore adjusted the wording (p.13, lines 374-376):

“As expected, only few of the investigated imaging parameters reached statistical significance, potentially due to the lower power resulting from the smaller group sizes than in the main analysis”

15. Table 2: should standardize format of scientific notation throughout. Also, explicitly state how many comparisons are included in the FDR correction. Based on the table, it seems FDR correction was only applied to the ANOVA comparison, while post-hoc testing of oral GC vs controls used uncorrected P-values? If so, it is not likely that $p=0.002$ would survive significance correction for the caudate..this will need clarification given it is highlighted throughout the paper if I understand the analysis correctly.

As explained above, for both primary and secondary analyses FDR correction was used. Subsequently, for those variables that were significant after FDR correction, a post-hoc analysis was performed. For the continuous parameters (imaging and emotional outcomes), this was done using Dunnett’s test. For the emotional parameters, that are expressed as odds ratios, Dunnett’s test was not possible, and post-hoc analysis was performed using Bonferroni correction for groupwise comparisons. This means that during the post-hoc testing, P values were corrected for multiple comparisons. We would like to refer again to the new figure in which this stepwise approach is visualized.

To clarify this in Table 2, we adjusted the legend (p.15, lines 409-410):

“ Adjusted mean difference, calculated using linear models, adjusted for age, sex, education, X-, Y-, and Z-position of the head in the scanner, head size, assessment centre, and year of imaging acquisition; significance was determined using a post-hoc Dunnett’s test.”*

We have also corrected the scientific notation in Table 2.

16. It is unclear why education is included in the adjusted means. I think it would be helpful to consider removing it if it does not add any scientific value to the model.

As addressed in our reply to comment 10, we have included education as a covariate because education is related to intelligence and cognitive functioning. We feel that the scientific value of adding this covariate to the model lies in the assumption that every difference in brain function potentially has a brain substrate.

17. Clarification is needed on the exclusion of outliers for the cognitive tests before the findings can be validly interpreted: were the outliers from the non-transformed data? Were outliers excluded iteratively for each measure tested or independently for each statistical test performed? Can sample size be included in the text for completeness?

For the cognitive tests, the outliers were removed after transformation of the data. As shown in Supplement 3.13, the outliers were determined independently per outcome variable per group. Because of this, the sample size differs for each measure tested. We feel it would add too much noise to the text if we mentioned all sample sizes in the main manuscript, but we do refer to Supplement 3.13.

For clarification, we added the following to the Statistical analysis section (p. 9, lines 305-306):

“For the cognitive parameters, the outliers were removed after transformation of the data.”

18. Are the cognitive/neuropsych tests standardized/scaled scores that adjust for age? Or is it raw scores used?

The test results are provided as raw scores by the UK Biobank. In our analyses, we adjusted them for age as well as sex and education.

19. Figure 2 has extremely small font in the bar graphs, making it difficult to read. Significance levels for group comparisons should be included in the bar chart.

We thank the reviewer for pointing this out. We have increased the font size and added the significance level to Figure 3 (previously Figure 2).

20. Graphical representations of numerical data, especially neuroimaging findings, would be advised to present in the form of a brain template if able, to better understand the neuroanatomic distribution of the findings. At the very least, scatter plots would be helpful to interpret the patterns in key findings.

We agree with the reviewer that the distribution of the data would be insightful to the readers. However, scatter plots are not possible for our data, because we have data in 3 categories, and

not continuous in a single x-y-plane. We therefore presented the data as violin plots in Supplement 4, which gives more insight into the number of data points observed per value. However, we decided to keep the bar plots in the main manuscript because these are more intuitive to understand.

Discussion

21. Expression and concentration of GRs in the brain have region specificity. Is there a way to further categorize the findings, even the counter-intuitive ones like increased caudate volume, with receptor expression profiles? Some discussion about the differential patterns of GR throughout key brain regions, like hippocampus/amygdala, PFC, etc., may be warranted to further illustrate and explain the results.

We thank the reviewer for this excellent suggestion. In a previous paper by our group (see reference below), we already tried to correlate the expression of GR and MR in several brain areas to the changes in brain volume observed in the extreme hypercortisolism caused by Cushing disease. We then concluded that, although a high expression of these receptors was seen in the key brain areas such as the hippocampus, anterior cingulate cortex, and amygdala, there was no straight-forward correlation between receptor expression profiles and brain areas affected by hypercortisolism. One might speculate that whether an area is affected by glucocorticoids may be more related to the densities of specific cell types that are responsive to glucocorticoids than the expression of receptors per se. Perhaps the density of oligodendrocytes, which are increasingly recognized as glucocorticoid-responsive, could be an important factor determining the responsiveness of different brain areas to glucocorticoids.

We have also added this to the discussion (p.22-21, lines 752-771):

“It is tempting to relate these findings to glucocorticoid (GR) and mineralocorticoid receptor (MR) expression profiles in the brain. Previously, our group correlated the expression of GR and MR in several brain areas (data from the Allen Brain Atlas ⁶⁹) to the changes in brain volume observed in the extreme hypercortisolism caused by Cushing disease. ²³ We then concluded that, although a high expression of these receptors was seen in the key brain areas such as the hippocampus, anterior cingulate cortex, and amygdala, there was no clear correlation between receptor expression profiles and brain areas affected by hypercortisolism. Receptor expression appears necessary but not predictive in this case. One might speculate that whether an area is affected by glucocorticoids may be more related to the densities of specific cell types that are responsive to glucocorticoids than the expression of receptors per se. Perhaps the density of oligodendrocytes, which are increasingly recognized as glucocorticoid-responsive, could be an important factor determining the responsiveness of different brain areas to glucocorticoids.”

Reference: Andela CD, van der Werff SJ, Pannekoek JN, van den Berg SM, Meijer OC, van Buchem MA, Rombouts SA, van der Mast RC, Romijn JA, Tiemensma J, Biermasz NR, van der Wee NJ, Pereira AM. Smaller grey matter volumes in the anterior cingulate cortex and greater cerebellar volumes in patients with long-term remission of Cushing's disease: a case-control study. Eur J Endocrinol. 2013 Oct 21;169(6):811-9. doi: 10.1530/EJE-13-0471. PMID: 24031092.

22. In the strengths/limitations portion, it is stated that psychiatric diseases like anxiety, depression, etc., were not excluded..but in the participants section of the method it states that they were excluded. It is a bit confusing to me how it is worded. Although it appears the rationale made explicit in the discussion states that the authors did not want to exclude patients based on potentially glucocorticoid-related outcomes, implying that these patients were not excluded, as they state here. Consider simplifying the language used in the methods section if that is the case..again, psychosis/schizophrenia is not deliberately stated amongst these conditions, which is curious and should be addressed as to why, however is included in the conditions listed in the supplemental.

We hope to have sufficiently addressed this question in our reply to comment 6.

23. In discussing potential confounds, it would be worthwhile to mention about the use of education as a covariate in analyses. Most of the cognitive/neuropsych testing has standardized scaled scores that do adjust for age already..but it's not clear if this is already done or the results are raw scores. Level of education may be a valid covariate in these analyses, but it is not known to me how conventional that is. May be helpful to note if it is going to be used in this specific analysis as well as in the analysis of the DTI and glucocorticoid relationships with and without education level as a covariate.

As stated above, the cognitive and neuropsychiatric data were provided as raw scores, not adjusted for age. Education is a rather conventional parameter used as potential confounder, both for cognitive outcomes and for imaging outcomes. Consider for example the following studies, that adjusted for education:

- Antal B, McMahon LP, Sultan SF, Lithen A, Wexler DJ, Dickerson B, Ratai EM, Mujica-Parodi LR. Type 2 diabetes mellitus accelerates brain aging and cognitive decline: Complementary findings from UK Biobank and meta-analyses. *Elife*. 2022 May 24;11:e73138. doi: 10.7554/eLife.73138. PMID: 35608247; PMCID: PMC9132576.
- Shen C, Rolls E, Cheng W, Kang J, Dong G, Xie C, Zhao XM, Sahakian B, Feng J. Associations of Social Isolation and Loneliness With Later Dementia. *Neurology*. 2022 Jun 8;10.1212/WNL.0000000000200583. doi: 10.1212/WNL.0000000000200583. Epub ahead of print. PMID: 35676089.
- Shen J, Tozer DJ, Markus HS, Tay J. Network Efficiency Mediates the Relationship Between Vascular Burden and Cognitive Impairment: A Diffusion Tensor Imaging Study in UK Biobank. *Stroke*. 2020 Jun;51(6):1682-1689. doi: 10.1161/STROKEAHA.119.028587. Epub 2020 May 11. PMID: 32390549; PMCID: PMC7610498.

Furthermore, it should be noted that if education were not a true confounder in our study, it also would not affect our outcomes. Education is not in the causal pathway and the number of participants is sufficiently large to prevent the models from becoming unstable. Also, we would like to emphasize that our models are not designed as prediction models.

24. The note that the limited sample size did not allow for significant findings in the chronic oral glucocorticoid group may be in error. Reference to other studies better powered for this specific claim may lend credence to it, however power alone was not assessed to make this determination. Authors can consider performing a power analysis to make such an inference in this subgroup.

For this remark, we would like to refer to our reply to the last remark about the Methods section. We have also adjusted the wording in the Discussion (p. 24, lines 839-840):

"...although the expected lower power of the small chronic systemic glucocorticoid group likely precluded most associations from reaching significance."

Conclusion

25. The association noted as a main conclusion is overly broad – 'decreased white matter integrity' should be more precisely defined based on region specificity the authors found to be most conclusive and important.

As described in the Methods section, the global FA and MD were calculated by averaging the FA and MD values over all white matter tracts. These were consistently associated with decreased FA and increased MD in glucocorticoid users, indicative of decreased white matter integrity. We also investigated several white matter tracts of interest that had previously been associated with decreased white matter integrity and found that most of them were significant associated with decreased FA and increased MD. However, because the global white matter integrity was decreased, as were most of the white matter tracts of interest, we do not feel that we can conclude that the investigated white matter tracts are the only or the most affected tracts. We cannot say anything about the tracts that were not investigated, but given the global changes, we expect that the other are affected to some extent as well. We therefore choose to not define the decreased white matter integrity more precisely, but to stress that we find an apparently global reduction in white matter integrity. Yet, we agree that we cannot bluntly assume the global nature of the white matter changes, and we made the following adjustments to the text.

“The most remarkable and consistent effects were observed in white matter integrity, as both systemic and inhaled glucocorticoid use was associated with widespread reductions in FA and increases in MD.” (p. 21, lines 708-710)

“This study shows that both systemic and inhalation glucocorticoids are associated with an apparently widespread reduction in white matter integrity...” (p. 25, lines 868-869)

Reviewer: 2

Dr. Konstantinos Kalafatakis, Panepistimio Ioanninon, University Hospital Heraklion

Comments to the Author:

This is a very interesting research effort, that utilizes UK Biobank data to study the effects of corticosteroid treatment (oral & inhaled) on MRI and behavioural/ psychological markers of brain structure/ function. This must be indeed the largest study to date assessing the effects of glucocorticoid on the brain, especially when it comes to inhaled steroids (which are very frequently used in COPD). My only remark would be the following:

I think the readers would benefit a lot by adding another Figure which illustrates step-by-step the (MRI and rest of) data processing pipeline, that was followed in each case. That way, the readers would see 3 Figures, one related to subject inclusion and numbers (already there), one on data curation, processing and analysis pipelines, and one with the results (already there).

We thank the reviewer for this helpful suggestion. We agree that a visual representation of the data analysis process would be insightful. We have therefore added Figure 2, in which we have visualized the statistical analysis pipeline. The imaging data processing pipeline was designed and performed by the UK Biobank, and we refer to their documentation in our manuscript.

VERSION 2 – REVIEW

REVIEWER	Goldwasser, Eric University of Maryland Baltimore, Psychiatry
REVIEW RETURNED	22-Jun-2022
GENERAL COMMENTS	Authors have satisfactorily addressed all concerns and comments raised. I thank them for their diligence, effort, and timeliness in doing so, and find the revised manuscript greatly improved.